# Retrospective evaluation of factors affecting successful fit testing of respiratory protective equipment during the early phase of COVID-19

Silvia Caggiari,[1] Dan Bader,[1] Zoe Packman,[2] Jane Robinson,[2] Sumeshni Tranka,[2] Dankmar Böhning,[3] Peter Worsley [ORCID] [1]

[1]School of Health Sciences, University of Southampton, Southampton, UK
[2]NHS England and NHS Improvement, London, UK
[3]Southampton Statistical Sciences Research Institute, University of Southampton, Southampton, UK

**Correspondence to**
Dr Peter Worsley;
p.r.worsley@soton.ac.uk

## ABSTRACT

**Objectives** Respiratory protective equipment is critical to protect healthcare workers from COVID-19 infection, which includes filtering facepiece respirators (FFP3). There are reports of fitting issues within healthcare workers, although the factors affecting fitting outcomes are largely unknown. This study aimed to evaluate factors affecting respirator fitting outcomes.

**Design** This is a retrospective evaluation study. We conducted a secondary analysis of a national database of fit testing outcomes in England between July and August 2020.

**Settings** The study involves National Health Service (NHS) hospitals in England.

**Participants** A total of 9592 observations regarding fit test outcomes from 5604 healthcare workers were included in the analysis.

**Intervention** Fit testing of FFP3 on a cohort of healthcare workers in England, working in the NHS.

**Primary and secondary outcome measures** Primary outcome measure was the fit testing result, that is, pass or fail with a specific respirator. Key demographics, including age, gender, ethnicity and face measurements of 5604 healthcare workers, were used to compare fitting outcomes.

**Results** A total of 9592 observations from 5604 healthcare workers were included in the analysis. A mixed-effects logistic regression model was used to determine the factors which affected fit testing outcome. Results showed that males experienced a significantly ($p<0.05$) higher fit test success than females (OR 1.51; 95% CI 1.27 to 1.81). Those with non-white ethnicities demonstrated significantly lower odds of successful respirator fitting; black (OR 0.65; 95% CI 0.51 to 0.83), Asian (OR 0.62; 95% CI 0.52 to 0.74) and mixed (OR 0.60; 95% CI 0.45 to 0.79.

**Conclusion** During the early phase of COVID-19, females and non-white ethnicities were less likely to have a successful respirator fitting. Further research is needed to design new respirators which provide equal opportunity for comfortable, effective fitting of these devices.

## STRENGTHS AND LIMITATIONS OF THIS STUDY

⇒ Results from a national audit of fit testing outcomes from healthcare workers using filtering facepiece respirators (FFP3).

⇒ Data revealed significant differences between genders and ethnicities in fit testing outcomes.

⇒ The study was limited to real-world data with no controls, during the early stages of the pandemic.

⇒ No direct comparisons between FFP3 respirators could be made.

⇒ Our secondary analysis reveals significant bias in fitting outcomes and the need for improvements in FFP3 respirator designs and standards.

## INTRODUCTION

The use of respiratory protective equipment (RPE) was vital in the prevention of airborne viral transmission for conditions such as influenza, SARS and SARS-CoV-2 (COVID-19). Indeed, a systematic review and meta-analysis demonstrated the use of N95 or filtering facepiece respirators (FFP3) respirators can reduce the risk of respiratory virus infection by 70%, suggesting respirator use offers significant protection against the transmission of such respiratory viruses.[1] In the specific context of COVID-19, the risk of infection was significantly reduced among healthcare workers (HCWs) wearing FFP3, highlighting the importance of RPE in the current pandemic.[2] As a result, there was an international effort to procure FFP3, often termed N95 respirators and FFP3 devices, creating an extreme demand on the associated supply chain.[3 4]

Fit testing is used to minimise the inward leakage of the respirator when attached to the face and both qualitative fit tests (QLFT) and quantitative fit tests (QNFT) have been recommended.[5 6] However, respirators with suboptimal fit are commonly used with a tightened strap used to create an airtight seal, thus endangering the health of the underlying skin on locations of the face.[7] Despite guidelines[8] and efforts to reduce skin damage,

the prevalence of reported issues has remained high.[9] There are many factors which may influence successful RPE fitting.[10] These include the relationship between facial dimensions and RPE shape and the materials used within the respirator which include both compliant and non-compliant designs.[11] Facial anthropometrics differences may influence the performance of RPE and there is already some evidence, although contradictory, that overall protection varies with gender-based differences in facial dimension.[12–15]

In England, the National Health Service (NHS) purchased a variety of FFP3 RPE devices to protect frontline HCWs in response to SARS-CoV-2 (COVID-19) as part of their pandemic preparedness stockpile. Recent studies in England have identified that male and White ethnicity HCWs are more likely to achieve RPE fit test success.[16 17] However, these were limited to a single healthcare centre limiting its generalisability. Thus, this study aims to analyse this data retrospectively and evaluate the fitting outcomes of a national population of HCWs using several variations of FFP3 respirators. Objectives of the study were to assess whether:

1. Intrinsic factors, namely, gender identity, ethnic background, age and body mass index (BMI) affected the success rates of FFP3 respirator fitting.
2. Specific respirator testing methods (qualitative or quantitative) affected the success rates.
3. Individual facial measurement parameters reflected the likelihood of successful respirator fitting.

## METHODOLOGY
This quality improvement study, involving NHS hospitals in England, collated data regarding fit test outcomes in all HCWs. Each healthcare provider and individual participant were consented to take part in the study.

### Fit testing data
Data included the success or failure of fitting or a range of FFP3 respirators (models A–G) according to two standardised and prescribed methods from the Health and Safety Executive, namely the QLFT or the QNFT.[18 19] This was conducted with a ToolKit provided (online supplemental e-Appendix A) to each hospital trust, through which details of demographics were collected, including gender, ethnic background, age and BMI. In the latter case, BMIs were categories using the WHO classifications as underweight ($<18.5 \, kg/m^2$), optimal ($18.5–25 \, kg/m^2$), overweight ($25–30 \, kg/m^2$) and obese ($>30 \, kg/m^2$).[20] In addition, five facial anthropometrical measurements were estimated with a paper-based ruler, namely, facial length, nasal length and protrusion, alar and biocular width. Inclusion criteria consisted of hospital staff working in areas deemed necessary for FFP3 respirators. HCWs were initially fitted with a respirator and assessed using the standardised tests. If they passed the respirator was retained, and those that failed had further attempts with other respirator models until successful (figure 1).

In some cases, HCWs were tested on multiple respirator variants even when successful fitting was initially achieved, this was conducted to counter any supply chain issues with specific respirator types.

Data collection was carried out in a 5-week period between July and August 2020. ToolKits were electronically distributed to a number of hospitals in England, identified with an anonymised code and responsible to name an experienced fit tester for the recruitment of staff members. All data were electronically collected, anonymised and stored securely within NHS England and Improvement.

### Data analysis
Data detailed the result of up to five fit test attempts for FFP3 respirator respirators in a cohort of >5000 hospital staff. These data were reviewed to ensure consistency in the analytical methods. This involved a cleansing process to minimise redundancy and remove outliers (from the paper-ruler facial measurements), missing or undeclared data. Subsequent review of the data included:

► Ethnic backgrounds initially categorised into 24 groups and subsequently into four major ethnic backgrounds, namely, white, Asian, black and mixed. Further analysis was conducted with respect to subgroups within each ethnicities associated with the specific country of origin.
► A few individuals identified their gender as 'non-binary', representing only 0.1% of the total cohort. These were excluded from the analysis, as they precluded an analysis of the effect of gender on success rates of respirator fitting.

### Statistical analysis
Cleaned data were imported into Stata (Stata V.16.0, StataCorp), where a basic model assessment was performed using the Bayesian information criterion (BIC). The model with the lowest BIC value was selected incorporating the fit test type, respirator model, gender, ethnicity and age. Following this evaluation, a mixed-effects logistic regression model was used to determine the ORs and their 95% CIs for factors which affected fit testing outcome using a collated data set across each of the five fit test attempts. Analysis was conducted while adjusting for multiple risk factors and potential confounders. This approach accounted for nested data (as several, different respirators were fitted to the same person) and for within-subject correlation by means of a random effect for subject.

Explorative data analysis was also conducted to evaluate trends between fitting outcomes within different ethnicities and gender and BMI categories. In addition, a cumulative frequency approach was used to investigate the distribution of the facial measurements. Cluster analyses were conducted to identify any trends in the measurements associated with fit test outcome.

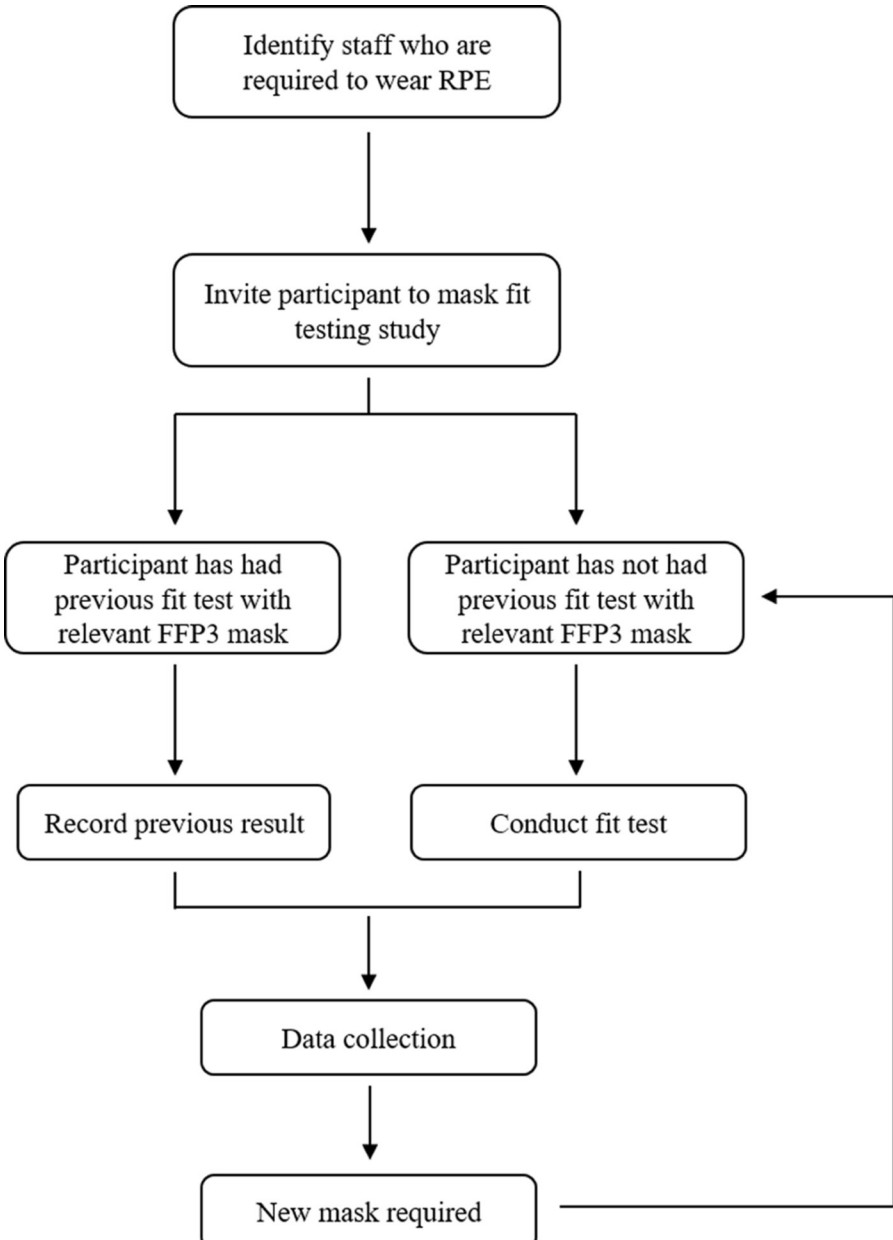

**Figure 1** Flow chart depicting the respirator testing process and repeat assessments which contributed to the data set. FFP3, filtering facepiece respirators; RPE, respiratory protective equipment.

## Patient and public involvement

Patients and public (HCWs) were consulted on the challenges associated with getting correctly fitting respirators and this formed the motivation for the quality improvement study. These were consulted regarding data collection forms and analysis.

## RESULTS
### Overview of the data

A total of 9592 observations from 5604 HCWs were included in the regression model (132 data entries were removed from the quality control process). This included data from 47 NHS Hospitals in England, where HCWs were fit tested with various respirator designs up to five times.

The number of fit attempts differed between individuals, with a steady decline from attempts 1 to 5. To review briefly, there were 5512, 2191, 1072, 560 and 257, in the first, second, third, fourth and fifth accumulative attempts, respectively (table 1). There was a small increase in the proportion of Asian individuals between the first (20.3%) and fifth (27.2%) attempts, while the gender and BMI categories remained largely unchanged. The majority of the individuals (~80% of the total for all attempts) were identified to be in the middle two BMI classifications. Individuals with a BMI under $18 \, \text{kg/m}^2$ represented the smallest proportion of the cohort (3%–5%) in all five

**Table 1** Summary of anthropometric data of participants of participants at each attempt of the fit test

|  | Total | Attempt 1 | Attempt 2 | Attempt 3 | Attempt 4 | Attempt 5 |
|---|---|---|---|---|---|---|
| No of fit tests | 9592 | 5512 | 2191 | 1072 | 560 | 257 |
| Successful fitting | 67.2% | 72.2% | 62.9% | 58.4% | 53.8% | 63.8% |
| Gender |  |  |  |  |  |  |
| Female | 79.1% | 79.6% | 78.9% | 78.6% | 77.0% | 74.7% |
| Male | 20.9% | 20.4% | 21.1% | 21.4% | 23.0% | 25.3% |
| Ethnicity |  |  |  |  |  |  |
| White (Caucasian) | 62.9% | 64.3% | 61.5% | 62.0% | 57.0% | 61.9% |
| Asian/Asian British | 22.3% | 20.3% | 23.1 % | 24.8% | 30.7% | 27.2% |
| Black/African/Caribbean | 8.4% | 8.9% | 8.8% | 6.8% | 6.6% | 5.8% |
| Mixed/multiple backgrounds | 6.4% | 6.5% | 6.6% | 6.3% | 5.7% | 5.1% |
| BMI |  |  |  |  |  |  |
| Under 18.5 kg/m$^2$ | 3.6% | 3.3% | 3.6% | 4.3% | 4.6% | 5.1% |
| 18.5–25 kg/m$^2$ | 51.3% | 50.2% | 51.0% | 53.3% | 57.0% | 58.0% |
| 25–30 kg/m$^2$ | 30.6% | 31.7% | 30.8% | 28.0% | 26.8% | 24.1% |
| Over 30 kg/m$^2$ | 14.5% | 14.9% | 14.6% | 14.4% | 11.6% | 12.8% |
| Age |  |  |  |  |  |  |
| 18–24 years old | 8.1% | 8.7% | 7.9% | 7.2% | 5.4% | 6.2% |
| 25–34 years old | 32.3% | 30.4% | 32.4% | 36.7% | 38.9% | 41.2% |
| 35–44 years old | 24.5% | 24.3% | 25.2% | 24.3% | 24.1% | 22.6% |
| 45–54 years old | 23.2% | 23.3% | 23.1% | 22.4% | 23.9% | 23.0% |
| 55+ years old | 11.9% | 13.2% | 11.3% | 9.4% | 7.7% | 7.0% |

BMI, body mass index.

attempts. Most of the individuals, approximately 78% of the total for all attempts, were aged between 24 and 54 years of age. Those that were younger (18–24 years) represented approximately 8% of the cohort, whereas those in the older categories (55+ years of age) represented approximately 12% of the participants.

### Mixed-effects logistic regression model
The final model after adjusting for covariates in our multilevel mixed-effects logistic regression analysis indicated that the odds of experiencing a successful FFP3 respirator fitting were significantly (p<0.05) higher among those who were male (OR 1.51; 95% CI 1.27 to 1.81) compared with females. In addition, those with non-white ethnicities demonstrated lower odds of successful respirator fitting, with black (OR 0.65; 95% CI 0.51 to 0.83), Asian (OR 0.62; 95% CI 0.52 to 0.74) and mixed (OR 0.60; 95% CI 0.45 to 0.79) ethnic groups all being statistically significantly (p<0.05) when compared with the white ethnic group. Individuals who were assessed with a QNFT had a significantly lower odds of successful fitting than those who performed a QLFT (OR 0.71; 95% CI 0.61 to 0.82). HCW with a low BMI (below 18.5 kg/m$^2$) had a significant lower odds of success fitting compared with all the other categories (table 2).

The likelihood ratio test (last line table 2) showed that the random effect for subject is strongly required, and the

simple logistic regression approach would have been not sufficient here.

### Explorative data analysis
Analysis of the 9592 observations showed that 71% of the fit tests were qualitative, as opposed to ~28% of QNFTs (table 3). Closer examination of the data revealed that the test type did not influence the relative proportion of success rates, with QNFT showing similar success and failure rates. In addition, the type of fit test performed was equally distributed across categories of gender and ethnicity.

Fit test outcomes also revealed that 60% of the HCWs were fit tested only once, as opposed to 4% who performed all the five attempts. In addition, data revealed that 17% of the individuals failed all the attempts performed, 60% passed one attempt, <15% of HCWs passed between 2 and 5 attempts.

With respect to ethnicity and gender, the group of white male participants yielded the highest success rates, with absolute differences of ≥4% compared with all other groups (table 4). In a similar manner, the white females yielded a higher success rate than their corresponding ethnic comparators, with a difference of ≥5%. Asian and mixed background ethnic groups yielded the lowest success rate of 60.3% and 60.4%, respectively. Closer examination of the fit test outcome also revealed the

**Table 2** ORs of a successful FFP3 respirator fit test outcome and the corresponding 95% CIs for each factor adjusted for all other factors using multilevel mixed-effects logistic regression

| Factor | | n | OR | z | P> \|z\| | 95% CI |
|---|---|---|---|---|---|---|
| Gender | Female | 7583 | 1 (base) | | | |
| | Male | 2009 | 1.514 | 4.61 | **0.000** | 1.269 to 1.806 |
| Ethnicity | White | 6034 | 1 (base) | | | |
| | Black | 810 | 0.622 | −5.36 | **0.000** | 0.523 to 0.741 |
| | Asian | 2133 | 0.653 | −3.40 | **0.001** | 0.511 to 0.835 |
| | Mixed | 615 | 0.603 | −3.54 | **0.000** | 0.456 to 0.798 |
| Type | Quantitative | 6801 | 1 (base) | | | |
| | Qualitative | 2725 | 0.7 | −4.71 | **0.000** | 0.619 to 0.821 |
| BMI | 18.5–25 | 4922 | 1 (base) | | | |
| | 25–30 | 2935 | 1.161 | 1.91 | **0.056** | 0.996 to 1.353 |
| | Over 30 | 1391 | 1.108 | 1.01 | **0.312** | 0.907 to 1.353 |
| | Under 18.5 | 344 | 0.516 | −3.66 | **0.000** | 0.362 to 0.735 |

LR test versus logistic model: chibar2(01)=−5680.43 Prob ≥chibar2=0.0000.
Bold values represent statistical significance (p<0.05).
BMI, body mass index; FFP3, filtering facepiece respirators; LR, likelihood ratio.

lowest success rate of approximately 56% associated with Bangladeshi HCWs, although they only represent a small proportion (54/2133, ~2.5%) of the corresponding Asian cohort.

### Fit test outcomes according to facial measurements

Facial measurements were examined with respect to the total cohort of 5544 HCWs. The cumulative frequency trends for each of the facial parameters were found to be normally distributed and, as a result, parametric descriptors were used for analysis. Close examination of the data suggests that there was a limited discrimination between facial measurements and fit test outcomes, with a wide range of values attributed to individuals who both passed

**Table 3** Summary of type of fit test across success rate, gender and ethnicity

| | Type of fit test | |
|---|---|---|
| | Qualitative | Quantitative |
| Total | 71.4% | 28.6% |
| Success rate | | |
| Passed | 69.4% | 61.8% |
| Failed | 30.6% | 38.2% |
| Gender | | |
| Female | 78.0% | 82.1% |
| Male | 22.0% | 17.9% |
| Ethnicity | | |
| White (Caucasian) | 63.2% | 61.2% |
| Asian/Asian British | 8.1% | 9.2% |
| Black/African/Caribbean | 22.2% | 22.3% |
| Mixed/multiple backgrounds | 6.5% | 5.9% |

and failed at each attempt (online supplemental e-Appendix B).

### DISCUSSION

A multicentre quality improvement study was implemented to explore the factors associated with fitting outcomes for FFP3 during the early phase of COVID-19 (July and August 2020). Conducted through NHS England and Improvement, data describing the fitting outcomes from multiple respirator models were completed for over 5000 HCWs. The multivariate analysis revealed that women and non-white ethnicities yielded a significantly lower OR of fit test success rates compared with males (table 2). Gender-based differences have been associated with fit testing outcomes in a number of studies, the majority of which demonstrate that female participants yield a significantly lower RPE success rate, and as a result need a range of respirator models for successful fitting.[12–16 21–28] This is in contrast to a recent study in the England, where no gender differences were identified.[15]

This study also observed clear ethnicity-based differences in fit test outcomes, both for males and females, with Asian, black and mixed ethic background individuals having a lower fit test success rate when compared with white ethnicities. White males yielded the highest success rate (74%) as opposed to 60% of Asian and mixed ethnicities females (tables 2 and 4). Small comparative studies have demonstrated lower pass rates for black and Asian females.[14 16 29 30] In addition, studies of Asian populations have consistently yielded higher rates of fit test failure within Chinese, Koreans, Taiwanese and Iranian cohorts, further emphasising the importance of considering facial dimensions of the relevant population in RPE design.[13 21 22 24–26 31–35] These differences are likely

Table 4  The absolute differences in global fit test success rate with reference to gender and ethnicity

| Gender and ethnic group (participants, % success rate) | White male | White female | Asian/Asian British male | Asian/Asian British female | Black/African/Caribbean male | Black/African/Caribbean female | Mixed/multiple backgrounds male | Mixed/multiple backgrounds female |
|---|---|---|---|---|---|---|---|---|
| White male (1046, 74.3%) | – | –5.8 | –4.4 | –14.0 | –3.5 | –10.8 | –7.6 | –13.9 |
| White female (4988, 68.5%) | | – | 1.4 | –8.2 | 2.3 | –5.0 | –1.8 | –8.1 |
| Asian/Asian British male (627, 69.9%) | | | | –9.6 | 0.9 | –6.4 | –3.2 | –9.5 |
| Asian/Asian British female (1506, 60.3%) | | | | – | 10.5 | 3.2 | 6.4 | 0.1 |
| Black/African/ Caribbean male (168, 70.8%) | | | | | – | –7.3 | –4.1 | –10.4 |
| Black/African/ Caribbean female (642, 63.5%) | | | | | | – | 3.2 | –3.1 |
| Mixed/multiple backgrounds male (168, 66.7%) | | | | | | | – | –6.3 |
| Mixed/multiple backgrounds female (447, 60.4%) | | | | | | | | – |

to have resulted from the known effects of ethnicity on anthropometric facial features.[36] Therefore, the RPE models currently available may not provide comparable protection in a multiethnic workforce, thus disadvantaging those from minority groups. This implies that RPE designs evaluated during this study did not accommodate the heterogeneity in facial features across diverse user populations due to the limited nature of the in panels used for international standards for example, EN-149.

It is also of note that fit testing success varied across the different BMI ranges, with the underweight category ($<18.5\,\mathrm{kg/m^2}$) yielding the highest failure rates. Closer analysis of the present findings also revealed that 40% of individuals with low BMI were Asian, 52% of whom did not achieve success at any attempt at fit testing. This suggests that both ethnicity related anthropometrics of face shape and BMI are contributing factors in fit test outcomes. Indeed, the current respirator fit panels do not accommodate those in the low BMI range, further evidence has shown that fitting will alter dependent on changes in an individual's weight.[37] Indeed, the soft tissue composition of the face could effect a change in the contact between the respirator and the underlying skin, with more bony prominences creating a less conforming surface from which respirators can create a seal.

This study has identified significant differences between the QNFT and QLFT success rates (table 2). This finding has also been demonstrated in other studies comparing the outcomes of these tests in cohorts evaluating the same respirator designs.[10] Here, there are a higher proportion of respirators passed using the qualitative methodology, and evidence that when a failure is observed in the quantitative test, a pass may still be achieved in the corresponding qualitative test for a given respirator.[38] Indeed, quantitative fit-testing has been defined as a gold standard and recommended to comply with international

and national standards.[10 38] There are some limitations with QLFT. First, it is a subjective test as it relies on the taste indicating the absence or presence of taste. In addition, the test hood may not be tolerated by HCWs with claustrophobia. It is also possible that some HCWs with increased anxiety may intentionally or unintentionally fail the fit test (indicating a leak) during a pandemic.[10]

This highlights the importance of an effective fitting process prior to use. Indeed, there is the need to support standard methods with intelligent fitting algorithms able to characterise the goodness of fit in an objective manner and predict respirator fitting, to provide individuals with safe and effective respirator.[39]

## Limitations

This secondary analysis of quality improvement data was limited by the nature in which individual hospital trust collected and documented fit testing. Indeed, a small proportion of data was omitted during the data cleaning process in order to conduct robust analysis. This was performed using defined criteria and enabled robust analysis, for example, removing mislabelled data or entries which did not include key parameters, for example, test type or ethnicity. The results of this retrospective analysis are only reflective of the time period of investigation (July–August 2020), with improvements to respirator designs and fitting processes evident during the latter stages of the pandemic. The data presented in this study are also a product of the international demands on PPE supply chain,[3] with a diverse range of respirators included in the data set. As a result, RPE in different English hospitals was variable and procurement dependent. In addition, implementing a comprehensive fit-testing programme is a financial and logistical challenge,[40] limiting the feasibility to test all HCWs on different FFP3 respirator models. Comparisons between respirators could not be

performed as there was no random allocation of respirator types across the fit testing attempts. The anthropometric data were limited by the use of paper-based rulers, with limited accuracy and reliability. Given the findings of gender and ethnicity differences, further evaluation of anthropometrics using accurate scanning technologies are required.

## Clinical implications

In practice, poorly fitted RPE designs impede both functional capacity and user safety.[41 42] Widespread concerns in areas of RPE fit-test access, availability and training have been highlighted during the pandemic.[43 44] Notably, for HCWs using RPE for prolonged periods, skin damage has been reported with a variable prevalence of between 42% and 97%. This has been attributed to ill-fitting RPE and may account for higher rates of adverse reactions within Ethnic minorities individuals.[9 45–47] In many countries, there have been significant improvements in the supply chain of RPE devices and a range of models is available to many healthcare institutions. Further research into the design and fitting of RPE must consider the demographics of the healthcare workforce including the diverse range of ethnicities, ages and genders. Design panels incorporating established data sets of different subgroups of demographics could inform new designs and international standards (eg, ISO/DIS 16976–6) by which FFP3 respirator s can be manufactured. Healthcare institutions should maintain a range of RPE devices to accommodate successful fitting for their diverse workforce and data shared with industrial representatives to ensure the available designs are fit for purpose.

## CONCLUSION

This secondary analysis of respirator fit testing outcomes revealed distinct trends in fit testing outcomes related to both gender and ethnicity. Indeed, white males were the most likely to yield a successful fit test outcome and ethnic minority females the least likely. In many cases, repeat fit testing with different respirator models was required to ensure individual staff had a correctly fitting device. Further research is needed to provide improvements in the fit testing approach resulting in a more intelligent selection process, and a more expansive selection of respirators is required to meet the diverse population of HCWs.

**Acknowledgements** We would like to thank all the participating hospital trusts support data collection during the Quality Improvement Initiative.

**Contributors** SC was responsible for conceptualisation, data acquisition, analysis, interpretation, writing original draft, review and editing. PW was responsible for conceptualisation, funding acquisition, supervision, writing critical review and editing. PW is the guarantor of the data analysis and interpretation. ZP, JR and ST were responsible for data acquisition for the study, constant coordination, writing critical review and editing. DBöhning was responsible for conceptualisation, funding acquisition, data acquisition, supervision, writing critical review and editing. DBader was responsible for statistical analysis and writing. PW will act as guarantor.

**Funding** This study was commissioned by NHS England and Improvement. The secondary analysis of data was supported by UK Research and Innovation (UKRI) as part of a funded project 'A Bio-Engineering approach for the SAFE design and fitting of Respiratory Protective Equipment (BE-SAFE RPE)' (Ref EP/V045563/1).

**Competing interests** None declared.

**Patient and public involvement** Patients and/or the public were involved in the design, or conduct, or reporting, or dissemination plans of this research. Refer to the Methods section for further details.

**Patient consent for publication** Consent obtained directly from patient(s).

**Ethics approval** Institutional ethics was approved by the Faculty of Environmental and Life Sciences (FELS) ethics committee to retrospectively analyse an anonymised version of the data set (ERGO-65166). Data were confidentially shared with the research group at the University of Southampton (UoS) and stored in accordance with the Data Protection Act 2018.

**Provenance and peer review** Not commissioned; externally peer reviewed.

**Data availability statement** Data are available on reasonable request. Technical appendix, statistical code and dataset available from the Pure repository, DOI:10.5258/SOTON/D2381.

**ORCID iD**
Peter Worsley http://orcid.org/0000-0003-0145-5042

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
