## [Reviewer comments · BMJ Open]

ARTICLE DETAILS

TITLE (PROVISIONAL)	Retrospective evaluation of factors affecting successful fit testing of Respiratory Protective Equipment during the early phase of COVID-19.
AUTHORS	Caggiari, Silvia; Bader, Dan; Packman, Zoe; Robinson, Jane; Tranka, Sumeshni; Böhning, Dankmar; Worsley, Peter

VERSION 1 – REVIEW

REVIEWER	Lytras, Theodore European University Cyprus
REVIEW RETURNED	20-Oct-2022

GENERAL COMMENTS	The manuscript has been substantially improved, and all my comments on its previous version have been addressed, most importantly (by far) the non-random allocation of mask types; this has now been appropriately acknowledged and the authors do not try to make inferences about that. As a result the paper is now much more focused, with the conclusions sufficiently supported by the data and analyses. Just one minor detail: page 8 line 217, "the youngest age category (18-24 years) had a significantly lower odds of success fitting compared to all older age categories (OR=2.55-3.45, $p<0.01$)". You can't have "lower odds" and $OR>1$ in the same sentence. Please rephrase (or use the inverse of this OR) so that the sentence makes sense. Thank you.
---

REVIEWER	Arora, Sumesh Prince of Wales Hospital, Intensive Care Medicine
REVIEW RETURNED	17-Dec-2022

GENERAL COMMENTS	Comments & Suggestions for change for BMJOpen-2022-065068 1 The terms mask and respirator have been used interchangeably in the article. I think only one term should be used, and 'respirator' is preferred to a 'mask'.2 Table 1 presents the success rate of the first attempt. It is of interest to the readers – How many participants eventually passed the fit test and in how many attempts? This information has not been provided. Please provide this information.3 Title of column II should say – 'Percentage of HCW Who Passed the fit test' Rather than 'Percentage of HCW fit tested'.4 The paper does not give any information about the qualitative fit tests (Using saccharine or a bitter solution as an indicator) vs Quantitative fit tests (e.g. using a machine like PortaCount). This should be provided. If the majority were tested using the qualitative test, the results may not be applicable for future comparisons as
--

	quantitative fit testing is set to take over qualitative fit testing. If the results of qualitative and quantitative fit testing are significantly different from each other, it may be argued that only the results of quantitative fit testing should be reported, o 5 'Success Rate' – The authors should indicate the success rate. I imagine that it is a 'Pass result on fit testing'. But it is not clear from the manuscript. None of the tables provides the 'success rate'. 6 Number of participants in each age bracket has not been provided. Please provide. 7 Please provide the 'n' for each respirator type in table 2 8 Number of participants who passed each respirator has not been provided. The authors should also indicate if the respirators were tested in a particular sequence. For example, if everyone was tested with 3M1863 first and with the others only if they failed, it would appear that 3M1863 is the best respirator. Or, were they tested with all respirators, regardless of the success rate (I doubt that this would be the case at the height of the pandemic). The odds ratio of success with each respirator depends on their relative position in the testing sequence. Let us say that we are testing 5 respirators. A, B, C, D and E. If a participant successfully fit tests the respirator A, then B-E are not tested. Now, imagine, in one hospital, the respirators are tested in the following sequence – A, B, C, D and E. In another hospital, they are tested in E, D, C, B and then A. Once a participant is successful in fit testing, the remaining respirators are not tested. The OR of successful respirator fit will be different in the two hospitals. Unless the sequence in which the respirators are fit tested, their data can not be combined. More information is required before the results of the OR of each respirator, as provided in Table 2, can be taken at face value. I note that the authors acknowledge this briefly in lines 317 to 321. It is my opinion that OR provided in table I may be assumed to be an indicator of relative respirator superiority by some readers. Some may interpret that 3M8833 is the 'best' respirator, which is not the appropriate conclusion. 9 With the above in consideration, there is insufficient substantiation for lines 48-49 in the abstract. Lines 48-49 of the abstract should be removed. 10 Line 277, 278 – The authors conclude that different respirators are independent predictors of success. Inadequate information has been provided to make this statement, and as such, the authors should not make that statement. What the study does prove is that each hospital should stock multiple types of respirators so that most HCWs may get the respirator that fits them the best.
--	---

REVIEWER	Gordon, Charlotte Northumbria University, Nursing, Midwifery and Health
REVIEW RETURNED	01-Feb-2023

GENERAL COMMENTS	This is an incredibly interesting and important study which adds to the body of knowledge of this topic and I was pleased to provide a review. Minor Comments: Abstract:  1. Specify context of the database within the abstract - 'national' - England 2. Ensure consistency in terminology with regard to devices under study - mask / respirators / respirator devices / filtering face-piece respirators. Specifically which masks are included in this analysis (N95 / FFP3) 3. 'Intelligent methods' is not clear in meaning - further detail suggested to ensure clarity for the reader
--

	4. Line 71 details the evaluation of N95 / FFP3 respirators in this study, line 120, line 214 and line 263 refer only to FFP3 masks - suggest consistent terminology here Introduction: 5. Given the time to publication since the pandemic suggest changing the tense of this section with regard to procurement of RPE. 6. Line 115 - change to 'recent studies in England have' (studies plural) 7. 118 - key questions should be presented as 'study objectives' with the overall aim clearly articulated / stated before Methodology: 8. Suggest consistency with regard to study objectives 'RPE model' revise to N95 / FFP3 9. Line 114 change RPE usage to N95 / FFP3 10. Figure 1 - should capture testing on multiple mask variants / fitting attempts (1-5 attempts) and could also reflect the numbers of participants at each stage of the study as this would summarise the process of data collection in the context of the quantity of data collected (participant numbers / numbers previously tested etc) Results: 11. Line 191 suggest change to 'the number of fit attempts' Discussion: 12. I would suggest that the structure of the discussion should reflect the key questions / study objectives identified in the introduction (factors affecting the success of fitting, variation in mask performance, mask model alignment to subject classification in terms of gender / ethnic group and affect of facial parameters) to show clear alignment between objectives and results / discussion, whilst these objectives are addressed in the discussion, the use of headings will assist the reader here. Conclusion: 13. I would suggest more detail on what is defined by an 'intelligent selection process' - evidence based?
--	---

VERSION 1 – AUTHOR RESPONSE

Reviewer: 1

1. page 8 line 217, "the youngest age category (18-24 years) had a significantly lower odds of success fitting compared to all older age categories (OR=2.55-3.45, p<0.01)". You can't have "lower odds" and OR>1 in the same sentence. Please rephrase (or use the inverse of this OR) so that the sentence makes sense. Thank you.

Response: We have rephrased the sentence as follow:

Line 229-230: In addition, the older age category (those 25 years and older) had a significantly higher odds of successful fitting compared to the youngest age category of 18-24 years (OR=2.55-3.45, p<0.01).

Reviewer: 2

1. The terms mask and respirator have been used interchangeably in the article. I think only one term should be used, and 'respirator' is preferred to a 'mask'.

Response: All the terms 'mask' has now been changed to 'respirator'.

2. Table 1 presents the success rate of the first attempt. It is of interest to the readers – How many participants eventually passed the fit test and in how many attempts? This information has not been provided. Please provide this information.

Table 1 has been updated for all fit testing attempts. In addition, we have detailed how many individuals failed all five attempts (17%).

Table 1 Modified to the following:

	Total	Attempt 1	Attempt 2	Attempt 3	Attempt 4	Attempt 5
Number of fit tests	9592	5512	2191	1072	560	257
Successful fitting	67.2%	72.2%	62.9%	58.4%	53.8%	63.8%
Gender						
Female	79.1%	79.6%	78.9%	78.6%	77.0%	74.7%
Male	20.9%	20.4%	21.1%	21.4%	23.0%	25.3%
Ethnicity						
White (Caucasian)	62.9%	64.3%	61.5%	62.0%	57.0%	61.9%
Asian/Asian British	22.3%	20.3%	23.1 %	24.8%	30.7%	27.2%
Black/African/Caribbean	8.4%	8.9%	8.8%	6.8%	6.6%	5.8%
Mixed/Multiple backgrounds	6.4%	6.5%	6.6%	6.3%	5.7%	5.1%
BMI						
Under 18.5 kg/m ²	3.6%	3.3%	3.6%	4.3%	4.6%	5.1%

18.5-25 kg/m ²	51.3%	50.2%	51.0%	53.3%	57.0%	58.0%
25-30 kg/m ²	30.6%	31.7%	30.8%	28.0%	26.8%	24.1%
Over 30 kg/m ²	14.5%	14.9%	14.6%	14.4%	11.6%	12.8%
Age						
18-24 years old	8.1%	8.7%	7.9%	7.2%	5.4%	6.2%
25-34 years old	32.3%	30.4%	32.4%	36.7%	38.9%	41.2%
35-44 years old	24.5%	24.3%	25.2%	24.3%	24.1%	22.6%
45-54 years old	23.2%	23.3%	23.1%	22.4%	23.9%	23.0%
55+ years old	11.9%	13.2%	11.3%	9.4%	7.7%	7.0%

3. **Title of column II should say – ‘Percentage of HCW Who Passed the fit test’ Rather than ‘Percentage of HCW fit tested’.**

Response: Title of column II has now been changed.

4. **The paper does not give any information about the qualitative fit tests (Using saccharine or a bitter solution as an indicator) vs Quantitative fit tests (e.g. using a machine like PortaCount). This should be provided. If the majority were tested using the qualitative test, the results may not be applicable for future comparisons as quantitative fit testing is set to take over qualitative fit testing. If the results of qualitative and quantitative fit testing are significantly different from each other, it may be argued that only the results of quantitative fit testing should be reported,**

Response: This has now been addressed. The type of fit test has been included in the Mixed-effects logistic regression model and detailed further in the exploratory analysis. Indeed, the data revealed that Individuals who were assessed with a quantitative fit test had a significantly lower odds of successful fitting than those who performed a qualitative fit test (OR = 0.71; 95% CI: 0.61 to 0.82).

5. **‘Success Rate’ – The authors should indicate the success rate. I imagine that it is a ‘Pass result on fit testing’. But it is not clear from the manuscript. None of the tables provides the ‘success rate’.**

Response: We confirm that the success rate is the pass result on fit testing. Table 3 states the percentage of success rate across ethnicities and genders. For clarity, we have amended the title of column II in Table 1 adding to read: ‘% success rate – First Attempt’.

6. **Number of participants in each age bracket has not been provided. Please provide.**

Response: Number of participants in each age range has been added.

7. **Please provide the ‘n’ for each respirator type in table 2**

Response: The respirator types have been removed according to previous comments regarding the limitations of our comparisons.

- 8. Number of participants who passed each respirator has not been provided. The authors should also indicate if the respirators were tested in a particular sequence. For example, if everyone was tested with 3M1863 first and with the others only if they failed, it would appear that 3M1863 is the best respirator. Or, were they tested with all respirators, regardless of the success rate (I doubt that this would be the case at the height of the pandemic). The odds ratio of success with each respirator depends on their relative position in the testing sequence. Let us say that we are testing 5 respirators. A, B, C, D and E. If a participant successfully fit tests the respirator A, then B-E are not tested. Now, imagine, in one hospital, the respirators are tested in the following sequence – A, B, C, D and E. In another hospital, they are tested in E, D, C, B and then A. Once a participant is successful in fit testing, the remaining respirators are not tested. The OR of successful respirator fit will be different in the two hospitals. Unless the sequence in which the respirators are fit tested, their data can not be combined. More information is required before the results of the OR of each respirator, as provided in Table 2, can be taken at face value. I note that the authors acknowledge this briefly in lines 317 to 321. It is my opinion that OR provided in table I may be assumed to be an indicator of relative respirator superiority by some readers. Some may interpret that 3M8833 is the ‘best’ respirator, which is not the appropriate conclusion.**

Response: The respirator types have been removed according to previous comments regarding the limitations of our comparisons.

- 9. With the above in consideration, there is insufficient substantiation for lines 48-49 in the abstract. Lines 48-49 of the abstract should be removed.**

Response: This has been removed.

- 10. Line 277, 278 – The authors conclude that different respirators are independent predictors of success. Inadequate information has been provided to make this statement, and as such, the authors should not make that statement. What the study does prove is that each hospital should stock multiple types of respirators so that most HCWs may get the respirator that fits them the best.**

Response: This has been removed.

Reviewer: 3

Abstract:

1. Specify context of the database within the abstract - 'national' - England

Response: This has been now specified. Lines 37-38: 'We conducted a secondary analysis of a national database of fit testing outcomes collated by a number of hospitals in England between July to August 2020'.

2. Ensure consistency in terminology with regard to devices under study - mask / respirators / respirator devices / filtering face-piece respirators. Specifically which masks are included in this analysis (N95 / FFP3)

Response: All the terms 'mask' have now been changed to 'respirator'.

The analysis includes FFP3 respirators, as highlighted in the introduction (Line 126). For clarity, we added this to the abstract: Line 51, to read: '...a variety of FFP3 respirator designs...'

3. 'Intelligent methods' is not clear in meaning - further detail suggested to ensure clarity for the reader

Response: Line 66: amended to read: '...Further research is needed to design new respirators which provide equal opportunity for comfortable, effective fitting of these devices. This will provide safeguarding for future pandemics and events where mass fitting of these devices is required.'

4. Line 71 details the evaluation of N95 / FFP3 respirators in this study, line 120, line 214 and line 263 refer only to FFP3 masks - suggest consistent terminology here

Response: Line 71 has been removed and all the respirators investigate in the study refer now to FFP3 models.

Introduction:

5. Given the time to publication since the pandemic suggest changing the tense of this section with regard to procurement of RPE.

Response: Thanks for your valuable comments. Amendments to the tense of the structure have been made, as follow:

Line 108 – Amended to read: 'The use of respiratory protective equipment (RPE) was vital...'

Line 127 – Removed to read: '...the National Health Service (NHS) has purchased...'

6. Line 115 - change to 'recent studies in England have' (studies plural)

Response: This has been now amended

7. 118 - key questions should be presented as 'study objectives' with the overall aim clearly articulated / stated before

Response: Thanks for your suggestion. We have now stated the overall aim and the study objectives, as follow:

Lines 131-141 – amended to read: Thus, the present study aims to analyse this data retrospectively and evaluate the fitting outcomes of a national population of healthcare workers using several variations of FFP3 respirators. Objectives of the study were to assess whether:

1. Intrinsic factors, namely, gender identity, ethnic background, age and BMI affected the success rates of FFP3 respirator fitting.
2. Specific respirator testing methods (qualitative or quantitative) affecting success rates
3. Individual facial measurement parameters reflected the likelihood of successful respirator fitting.

Methodology:

8. Suggest consistency with regard to study objectives 'RPE model' revise to N95 / FFP3

Response: Line 153 – Amended to read: 'Data included the type and make of FFP3 respirators used and...'

9. Line 114 change RPE usage to N95 / FFP3

Response: Line 162 – 'RPE usage' has been changed with 'FFP3 respirators.'

10. Figure 1 - should capture testing on multiple mask variants / fitting attempts (1-5 attempts) and could also reflect the numbers of participants at each stage of the study as this would summarise the process of data collection in the context of the quantity of data collected (participant numbers / numbers previously tested etc)

Response: Thanks for your valuable suggestion. Figure 1 reflects data collection. We retrospectively analysed the data when collection was completed, therefore data such as the numbers of participants at each attempt, numbers previously tested are part of the results and not the methodology.

Results:

11. Line 191 suggest change to 'the number of fit attempts'

Response: Line 211 (ex 192) – This has been amended accordingly.

Discussion:

12. I would suggest that the structure of the discussion should reflect the key questions / study objectives identified in the introduction (factors affecting the success of fitting, variation in mask performance, mask model alignment to subject classification in terms of gender / ethnic group and affect of facial parameters) to show clear alignment between objectives and results / discussion, whilst these objectives are addressed in the discussion, the use of headings will assist the reader here.

Response: The discussion has been modified accordingly. Now with sections on factors affecting fitting (gender and ethnicity). Impact of fit testing type (Qual vs Quant), limitations and clinical implications.

Conclusion:

13. I would suggest more detail on what is defined by an 'intelligent selection process' - evidence based?

Response: The final section of the conclusion has been modified to read 'Further research is needed to provide improvements in the fit testing approach resulting in a more intelligent selection process, and a more expansive selection of respirators is required to meet the diverse population of healthcare workers.'